# Region-Specific Sialylation Pattern of Prion Strains Provides Novel Insight into Prion Neurotropism

**DOI:** 10.3390/ijms21030828

**Published:** 2020-01-28

**Authors:** Natallia Makarava, Jennifer Chen-Yu Chang, Ilia V. Baskakov

**Affiliations:** 1Center for Biomedical Engineering and Technology, University of Maryland School of Medicine, Baltimore, MD 21201, USA; nmakarava@som.umaryland.edu (N.M.); cchang1@som.umaryland.edu (J.C.-Y.C.); 2Department of Anatomy and Neurobiology, University of Maryland School of Medicine, Baltimore, MD, 21201, USA

**Keywords:** prions, prion disease, N-linked glycans, sialic acid, sialylation, prion strains, thalamus, two-dimensional gel electrophoresis

## Abstract

Mammalian prions are unconventional infectious agents that invade and replicate in an organism by recruiting a normal form of a prion protein (PrP^C^) and converting it into misfolded, disease-associated state referred to as PrP^Sc^. PrP^C^ is posttranslationally modified with two N-linked glycans. Prion strains replicate by selecting substrates from a large pool of PrP^C^ sialoglycoforms expressed by a host. Brain regions have different vulnerability to prion infection, however, molecular mechanisms underlying selective vulnerability is not well understood. Toward addressing this question, the current study looked into a possibility that sialylation of PrP^Sc^ might be involved in defining selective vulnerability of brain regions. The current work found that in 22L -infected animals, PrP^Sc^ is indeed sialylated in a region dependent manner. PrP^Sc^ in hippocampus and cortex was more sialylated than PrP^Sc^ from thalamus and stem. Similar trends were also observed in brain materials from RML- and ME7-infected animals. The current study established that PrP^Sc^ sialylation status is indeed region-specific. Together with previous studies demonstrating that low sialylation status accelerates prion replication, this work suggests that high vulnerability of certain brain region to prion infection could be attributed to their low sialylation status.

## 1. Introduction

Mammalian prions are unconventional infectious agents that consist of misfolded, self-replicating states of a sialoglycoprotein called the prion protein or PrP^C^ [1,2]. Prions replicate by recruiting and converting PrP^C^ molecules expressed by a host into misfolded, self-replicating states referred to as PrP^Sc^ [3,4]. While prions are unconventional pathogens, they spread from cell to cell in CNS and elicit neuroinflammatory response that resembles the response of CNS to viral infections [5,6,7]. Moreover, like diseases caused by conventional agents, prion diseases can be transmitted between hosts via natural routes. In a striking resemblance of strain phenomenon of viral and microbial pathogens, multiple strains of prions or PrP^Sc^ that invade and replicate within the same host species were identified [8]. Prion strain of natural origin including mouse strains used in the current study were originally isolated from animals that succumbed to prion disease and then adapted to rodents via serial passaging [9,10,11]. Different strains elicit different, strain-specific disease phenotypes, characterized by strain-specific incubation time to disease, tropism to different brain areas and strain-specific tropism to different cell types [7,8,12]. The diversity of disease phenotypes within the same host is attributed to the ability of PrP^C^ to acquire multiple, alternative, conformationally distinct, self-replicating PrP^Sc^ states or strains [13,14,15,16,17,18]. Indeed, a number of studies provided solid evidence that that prion strains are different with respect to their biochemical properties as well as secondary, tertiary and quaternary structure [17,19,20,21,22,23]. While structural diversity of PrP^Sc^ strains has been well documented [19,20,21], the questions why and how multiple PrP^Sc^ structures, formed within the same amino acid sequence, elicit multiple disease phenotypes or target different brain areas remains poorly understood.

Two major types of posttranslational modifications, an attachment of the GPI anchor and N-linked glycosylation, were found in PrP^C^ [24,25,26]. The majority of PrP^C^ is diglycosylated (up to 80%), whereas a small fraction is monoglycosylated and very minor amount is unglycosylated [27]. In PrP^C^ N-linked glycans, sialic acids are terminal residues that are linked to galactose via α2-3 or α2-6 linkages with the majority being linked via α2-6 linkage [25,28,29]. More than 400 different PrP^C^ glycoforms have been identified, the heterogeneity attributed to the variations in structure and composition of the N-linked glycans [25,28]. The GPI-anchor and N-linked glycans are preserved upon conversion of PrP^C^ into PrP^Sc^ [30,31,32]. 

Upon conversion of PrP^C^ into PrP^Sc^, N-linked glycans are positioned on a surface of PrP^Sc^ particles and impose considerable spatial constrains to PrP^Sc^ assembly due to their bulky size and electrostatic repulsion between sialic acid residues [33,34]. In some strains, sialoglycoforms are recruited proportionally to their representation in PrP^C^ [27,33]. However, there are strains that exhibit a selectivity. Such strains preferentially recruit monoglycosylated and moderately sialylated PrP^C^ molecules at the expenses of diglycosylated and highly sialylated PrP^C^ glycofoms, which helps to overcome spatial and electrostatic restrains [27,33]. Strain-specific selection of PrP^C^ sialoglycoforms produces strain-specific patterns of carbohydrate epitopes on the surface of PrP^Sc^ [35].

Are carbohydrate groups on PrP^Sc^ surfaces important in eliciting biological response? The innate immune system is believed to sense terminal carbohydrate moieties including galactose and sialic acid residues. These groups are likely to serve as molecular cues and can trigger diverse responses by glia [36,37,38,39,40]. Sialic acid residues, which are abundant on the surfaces of all mammalian cells, act as a part of “self-associated molecular pattern” helping cells of the innate immune system including microglia to recognize “self” from “altered self” or “non-self” [41,42]. “Eat me” signals for professional and non-professional macrophages could be generated by galactose exposed on a cell surface upon removal of sialic acid residues [43,44]. As terminal residues, sialic acids are positioned on a surface of PrP^Sc^ particles, and are accessible for intermolecular interactions [34]. Recent studies from our laboratory revealed that sialylation of PrP^Sc^ N-linked glycans plays an important role in controlling prion fate in an organism [33,45,46,47,48]. Donor PrP^Sc^ with reduced sialylation levels did not induce prion disease in animals upon intracranial or peripheral administration [45,47,48]. Moreover, prion infectivity could be switched off and on in a reversible manner via removing and reinstalling sialylation of PrP^Sc^, respectively [47]. Sialylation status of PrP^Sc^ was also found to be important for prion lymphotropism [46,48]. Upon infection via peripheral route, PrP^Sc^ with normal sialylation status was sequestered by spleen and lymph nodes, whereas partially desialylated PrP^Sc^ was targeted predominantly to the liver [48]. Together, these studies suggested that sialylation protects PrP^Sc^ against clearance and appears to be critical in controlling the fate of prion infection in an organism.

Prion strains are known to invade brain regions in a strain-specific manner, a phenomenon referred to as selective neurotropism [49,50]. Closely related to this phenomenon is selective vulnerability of brain regions to prion infection, where some regions are being affected more severely than others. Molecular mechanisms underlying selective vulnerability are not well understood. It is not known whether PrP^Sc^ sialylation is involved in defining differential response of brain region to prion infection. While high sialylation levels of PrP^Sc^ appears to be critical for prion survival on the one hand [45,47,48], a decrease in sialylation level of PrP^C^ increase prion replication rate on the other hand [27,45]. It is not known whether strain-specific sialylation pattern of PrP^Sc^ is maintained across all brain region or whether region-specific differences exists. The current work demonstrated that PrP^Sc^ is sialylated in a region-dependent manner. Similar trends were observed in brain materials from 22L-, RML- and ME7-infected animals. The current study established that PrP^Sc^ sialylation status is indeed region-specific. Together with previous studies demonstrating that low sialylation status accelerates prion replication, this work suggests that high vulnerability of certain brain region to prion infection is attributed to their low sialylation status.

## 2. Results

### 2.1. Strain-Specific Sialylation Patterns of PrP^Sc^

To illustrate strain-specific sialylation patterns of PrP^Sc^, scrapie brain homogenates from animals infected with three mouse-adapted prion strains (22L, ME7 and RML) were analyzed using two-dimensional (2D) gel electrophoresis (Figure 1) [51]. Prior to 2D gel electrophoresis, the samples were denatured, so that the sialylation status of individual PrP molecules could be visualized as a distribution of charge isoforms in the horizontal dimension of 2D [51]. Since each sialic acid residue adds negative charges to individual PrP molecules, hypersialylated PrP molecules run toward acidic pH whereas hyposialylated toward basic pH. Three horizontal rows of charged isoforms corresponded to the non-, mono- and diglycosylated PrP molecules. Consistent with previous studies [27,46,52,53], non-glycosylated PrP molecules showed multiple charge isoforms on 2D (Figure 1). The structural heterogeneity of the GPI anchors, which could be also sialylated, account for this charge heterogeneity [54,55]. In RML material, monoglycosylated sialoglycoforms dominated over the diglycosylated sialoglycoforms, whereas diglycosylated glycoforms were predominant in both ME7 and 22L. Moreover, the hyposialylated isoforms were populated at a higher level in RML versus 22L and ME7. 22L and ME7 displayed very similar profiles of sialoglycoforms, nevertheless, ME7 showed the lowest ratio of non- and monoglycosylated versus diglycosylated glycoforms among the three strains (Figure 1). 

In summary, 2D analysis demonstrated subtle, yet notable differences between sialylation patterns of PrP^Sc^ of three strains. Next, we asked whether strain-specific sialylation patterns are preserved across brain regions. As an alternative possibility, PrP^Sc^ sialylation patterns might change with a region. If this is the case, the strain-specific sialylation pattern of PrP^Sc^ of a whole brain represents a weighted average of the region-specific PrP^Sc^ sialylation patterns. 

### 2.2. Region-Specific Sialylation of PrP^Sc^

For examining region-specific sialylation patterns, four brain regions including cortex, hippocampus, thalamus and stem from animals infected with 22L PrP^Sc^ were analyzed using 2D. Preliminary studies revealed that by the terminal stage of the diseases, 22L PrP^Sc^ accumulated throughout the brain (Figure 2A–K). Western blotting showed comparable amounts of PrP^Sc^ in cortex, hippocampus, thalamus and stem (Figure 2A). Immunohistochemistry confirmed PrP^Sc^ deposition in all four brain regions, and revealed several types of PrP^Sc^ aggregates including diffuse and punctate deposits and small plaques (Figure 2B–K). 

Comparison of cortex, hippocampus, thalamus and stem on 2D revealed differences in relative intensities of individual sialoglycoforms, which appeared to be subtle upon initial observation (Figure 3A). For visualizing the difference in more details, we applied artificial color assignment using Alpha View software, and analyzed three series of glycoforms (di-, mono and unglysocylated) separately (Figure 3B). Despite variations between animals, brain regions in individual animals displayed the same ranking order with respect to levels of sialylation within populations of di- and monoglycoylated glycoforms (from the most hypersialylated to hyposialylated): hippocampus = cortex > thalamus > stem (Figure 3B,C). In contrast to di- and monoglycosylated isoforms, the distribution of non-glycosylated charged isoforms, that lacked N-glycans, was very similar across brain regions, confirming that the differences in charge isoform distributions within di- or mono-glycoforms are attributed to N-linked glycans (Figure 3B,C). These results suggest that the sialylation of PrP^Sc^ N-glycans varies in a region-specific manner and that in hippocampus and cortex PrP^Sc^ was sialylated at higher levels than in thalamus and stem. 

For testing whether the ranking order of PrP^Sc^ sialylation is strain-specific or universal, we analyzed brains of animals infected with two additional mouse strains RML and ME7. Because only few brains were available for 2D analysis, statistical differences could not be established. Nevertheless, brain regions from RML and ME7 animals showed the same ranking order with respect to the sialylation levels as 22L-infected animals (Figure 4A,B). Together, these data suggest that sialylation levels of PrP^Sc^ are region-dependent. 

### 2.3. Region-Specific Expression of Sialyltransferases

In a cell, steady state sialylation status of glycoproteins is determined by two groups of enzymes: sialyltransferases (ST), which transfer sialic acid residues to glycans, and neuraminidases or sialidases, which cleave sialic acid residues [56,57]. Our previous studies demonstrated that knocking down of sialidases that expressed in CNS (*Neu1*, *Neu3*, *Neu4* or *Neu3/Neu4* double knockouts) did not affect the steady-state sialylation levels of PrP^C^ [58]. Among a large class of mammalian STs, five STs display a substrate specificity for sialylating N-linked glycans via α2-3 or α2-6 linkages, the type of linkages identified in PrP^C^ and PrP^Sc^ [25,28,29]. Three STs of the ST3 family (ST3Gal3, ST3Gal4 and ST3Gal6) sialylate via α2-3 linkages, whereas two STs of the ST6 family (ST6Gal1 and ST6Gal2) use α2-6 linkages [57,59].

For testing whether region-specific differential sialylation of PrP^Sc^ could be attributed to the region-specific expression level of STs, mRNA expressions of *ST3Gal3, ST3Gal4, ST3Gal6, ST6Gal1* and *ST6Gal2* were analyzed in hippocampus, cortex, thalamus and stem using qRT-PCR. In addition, we were interested in finding out whether different brain regions have differential expression of these STs under normal condition, and whether their expression changes with the prion disease. *ST3Gal3, ST3Gal4* and *ST3Gal6* levels of expression did not show reliable differences between brain regions neither in normal nor in infected animals (Figure 5A). *ST6Gal1* was the only gene that showed a significant increase in the prion-infected animals versus normal age-matched controls (Figure 5B), which was consistent with previous observations [60]. Another α-2,6-syaliltransferase, *ST6Gal2*, was expressed at much lower levels than *ST6Gal1* when compared to the housekeeping gene, yet was the only one displaying moderate region-specific differences in the expression level (Figure 5B). The region-specific differences in mean *ST6Gal2* expression values, while showing a higher expression levels in cortex relative to other regions, did not prove to be statistically significant. However, the *ST6Gal2* expression in individual animals consistently followed a similar pattern where the cortex displayed the highest level of *ST6Gal2* expression (Figure 5C). While analysis of STs expression does not explain relative ranking of brain region with respect to PrP^Sc^ sialylation, these data suggest that *ST6Gal2* might be responsible in part for region-specific differences in sialylation of PrP^Sc^.

## 3. Discussion

Recent discovery highlighted an important role of PrP^Sc^ sialylation in determining the rate of prion replication and their fate in an organism, shedding a new light on prion pathogenesis [27,33,40,45,46,47,48]. Considering that prion deposition in brain selectively targets specific regions and that brain regions display differential vulnerability to prion infection [7,8,12], we decided to look into the possibility that sialylation of PrP^Sc^ might be important in defining selective vulnerability. Consistent with this hypothesis is a previous work illustrating that in mice with deficient PrP^C^ glycosylation regional distribution of PrP^Sc^ and lesion profile underwent significant changes [61]. Within the current study, we asked the question whether PrP^Sc^ sialylation is uniform across brain regions or region-specific. Analysis of 22L-infected animals revealed that PrP^Sc^ is sialylated in a region-specific manner. This observation is consistent with the hypothesis in that a link between PrP^Sc^ sialylation and differential vulnerability exist. 

Despite variations in PrP^Sc^ sialylation pattern between individual 22L-infected animals, the common trend emerged upon analysis of this animal group. PrP^Sc^ deposited in hippocampus and cortex was found to be more sialylated than PrP^Sc^ from thalamus and stem. Similar trends were observed in brains from RML- and ME7-infected animals: PrP^Sc^ from cortex and hippocampus was more sialylated than PrP^Sc^ from stem. For RML and ME7 animals, statistically sound differences between brain regions could not be established, because only two and one animals, respectively, from those groups were available for analysis. Nevertheless, based on 22L data, this work indicates that in addition to strain-specific differences, PrP^Sc^ sialylation pattern is controlled by a brain region. 

The regional differences in sialylation status of PrP^Sc^ are possibly attributed to regional differences in sialylation of PrP^C^, a substrate of PrP^Sc^ replication. The fact that all three prion strains show similar trends supports this idea. Unfortunately, assessing sialylation status of PrP^C^ was proven to be very difficult due to uncanonical behavior of octarepeat region in isoelectric focusing [45,51,58]. In PrP^Sc^, the octarepeat is cleaved by proteinase K prior to 2D, which result in a well-defined positioning of individual sialoglycoform on the horizontal dimension of 2D [51]. 

What enzymes control sialylation status of N-linked glycans? In our previous studies, knocking down CNS sialidases in animals (*Neu1*, *Neu3*, *Neu4* or *Neu3/Neu4* double knockouts) did not change the steady-state sialylation levels of PrP^C^ [58]. In the current work, we tested whether region-specific differences in sialylation of PrP^Sc^ correlate with regional expression of those STs that sialylate N-linked glycans via α2-3 or α2-6 linkages (*ST3Gal3, ST3Gal4, ST3Gal6, ST6Gal1* and *ST6Gal2).* All three ST3s tested (*ST3Gal3, ST3Gal4, ST3Gal6)* displayed large variations between individual animals, but no consistent correlation with the regional sialylation levels of PrP^Sc^. *ST6Gal1* showed a statistically significant increase in expression levels in 22L-infected versus age-matched control animals at least in two brain areas, which was consistent with previous studies [60]. However, no correlation between regional *ST6Gal1* expression and sialylation levels of PrP^Sc^ was observed. The data on expression of STs mRNAs should be considered with a great caution, because mRNA level might not reflect the level of sialyltransferase enzymatic activity. 

Notably, expression of *ST6Gal2* was higher in cortex, a region with high sialylation status of PrP^Sc^, relative to hippocampus, thalamus and stem. While *ST6Gal1* gene is expressed in almost all human tissues, *ST6Gal2* shows a restricted tissue-specific pattern and mostly expressed in embryonic and adult brain [62]. Nevertheless, the substrate specificity of ST6Gal2 enzyme is not well understood. Recombinant ST6Gal2 was found to exhibit high sialyltransferase activity toward oligosaccharides; however, its activity toward glycoproteins was very low [63,64]. Several transcriptional activators and repressors of *ST6gal2* gene including NF-κB repressor were identified suggesting that *ST6gal2* transcription is regulated in a function dependent manner. The current work suggests that region-specific differences in sialylation of PrP^Sc^ might be attributed, at least in part, to differential expression of *ST6Gal2* in brain regions. It would be interesting to test the effect of *ST6Gal2* knock out on PrP^Sc^ sialylation and prion propagation. Differences in region-specific stability of hypersialylated versus hyposialylated PrP^C^ offers an alternative hypothesis for explaining region-specific differences in sialylation of PrP^Sc^. The third range of possibilities involves region-specific differences in N-glycan structures. Structural differences might be attributed to (i) regional differences in the ratios of bi- vs. three- and tetra-antennary N-glycan branches leading to a different number of potential sialylation sites for each glycan, and/or (ii) the degree of galactosylation of each antennary termini, as lack of galactosylation would prevent subsequent sialylation. The last mechanism was found to control sialylation status of N-glycans on IgG [65].

Do brain areas respond differently to prion infection? Why are some brain regions more vulnerable to prion infection than others? Recent studies illustrated that astrocytes respond to prion infection in a region-specific manner [66,67]. Transcriptome analysis and single-cell RNA-sequencing demonstrated great region-specific heterogeneity in phenotypes of microglia and astrocyte under normal conditions as well as revealed dynamic transformation of their phenotypes under aging and neurodegenerative diseases [68,69,70,71,72,73]. Expression of genes associated with neuroinflammation, synapse elimination and neuronal damage were found to be elevated in normal aging [70,71,72]. Notably, the rates of astrocyte aging under normal conditions vary as a function of brain region [72]. Considering that the mean age of onset of sporadic CJD is 66 years old, in humans the age-related changes in glia should be taken into consideration. However, the extent to which age-related changes in microglia or astrocytes contribute to region-specific susceptibility to prion infection in mice is questionable, because animal progress to terminal stage much earlier than the age-related changes take place. 

In 22L-infected animals, the thalamus display prion deposition and chronic neuroinflammation prior to cortex and hippocampus [7,74]. Moreover, by the terminal stage of the disease, the thalamus is affected more severely than other brain regions [7,74]. In mice, high vulnerability of thalamus to prion infection was found to be irrespective of a prion strain [7,12,66,74,75]. What, then, makes the thalamus so vulnerability to prion invasion and chronic neuroinflammation? The current study demonstrated that PrP^Sc^ from thalamus and stem had lower sialylation levels relative to PrP^Sc^ from cortex or hippocampus (Figure 3). Considering that low sialylation status accelerates prion replication [27,45], the results of the current work suggest that thalamus and stem are more vulnerable to prion infection due to its low sialylation status. In addition to plausible region-specific differences in replication rates, sialylation status of PrP^Sc^ might impact the severity of pathogenesis via controlling the pro-inflammatory response of microglia. Results of our recent study provide experimental support behind this hypothesis [40]. Partial desialylation of PrP^Sc^, which results in an increase in the number of galactose residues at the terminal position of PrP^Sc^ N-linked glycans, was found to boost the pro-inflammatory response of microglia [40]. Together with previous work, the current study suggests that region-specific differences in sialylation might be important factor in controlling the timing and severity of prion pathogenesis. 

Since PrP^Sc^ sialylation patterns is determined to large extent by a region, the strain-specific sialylation of PrP^Sc^ of a whole brain represent a weighted average of the region-specific PrP^Sc^ sialylation patterns for each individual strain. We propose that strains that can tolerate hypersialylated PrP^C^ molecules as a substrate are better positioned to spread into brain regions expressing hypersialylated PrP^C^ relative to the strains that are more selective with respect to PrP^C^ sialylation status. Whether this is the case will have to be determined in future studies. 

## 4. Materials and Methods 

### 4.1. Ethics Statement

This study was carried out in strict accordance with the recommendations in the Guide for the Care and Use of Laboratory Animals of the National Institutes of Health. The animal protocol was approved by the Institutional Animal Care and Use Committee of the University of Maryland, Baltimore (Assurance Number A32000-01; Permit Number: 0118001).

### 4.2. Animals

C57BL/6J mice (females and males) were inoculated intracerebrally into the left hemisphere ~2 mm to the left of the midline and ~2 mm anterior to a line drawn between the ears with 20 μL of 1% 22 L brain homogenate under isoflurane anesthesia. Inoculum is delivered slowly by a 26 G needle inserted to a depth of approximately 3 mm. Signs of neurological disease were detected between 132–138 days post inoculation and consisted of hind-limb clasp, ataxia and weight loss. Within 13–23 days after first clinical sings, mice became unable to walk on a beam, developed kyphosis and became lethargic. Mice were considered terminally ill when they were unable to rear and/or lost 20% of their weight. At this point they were euthanized by CO_2_ asphyxia and decapitation.

### 4.3. Histopathology

Formalin-fixed brain halves divided at the midline (left hemispheres) were treated in formic acid (95%) to deactivate prion infectivity before being embedded in paraffin. 4 µm sections mounted on slides were processed for immunohistochemistry. To expose PrP^Sc^ epitopes, slides were subjected to 20 min hydrated autoclaving at 121 °C in trisodium citrate buffer, pH 6.0 with 0.05% Tween 20, followed by 5 min treatment with 88% formic acid. PrP was stained with anti-prion antibody SAF-84 (Cayman Chemical, Ann Arbor, MI, USA).

### 4.4. 2D Electrophoresis

Brains were divided at the midline, and right hemispheres were used to dissect cortex, hippocampus, thalamus and stem. Each brain region was further divided into two pieces: one for protein electrophoresis, and another one for qRT-PCR (see below). 10% (wt/vol) homogenates from each brain region were prepared within 1.5 mL tubes in ice-cold PBS, pH 7.4, using RNase-free disposable pestles (Fisher scientific, Hampton, NH, USA). To prepare samples for 2D electrophoresis, 10% homogenates were diluted with 9 volumes of 1% Triton X-100 in PBS, pH 7.4, and digested with 20 µg/mL PK for 30 min at 37 °C. The reaction was stopped by addition of NuPAGE LDS sample buffer (Thermo Fisher Scientific, Waltham, MA, USA) and a subsequent 10 min incubation in boiling water bath. After that, 25 µL of samples in sample buffer were solubilized for 1 h at room temperature in 200 µL solubilization buffer (8 M Urea, 2% (wt/vol) CHAPS, 5 mM TBP, 20 mM TrisHCl pH 8.0), then alkylated by adding 7 µL of 0.5 M iodoacetamide and incubation for 1 h at room temperature in the dark. Then, 1160 µL of ice-cold methanol was added and the samples were incubated for 2 h at −20 °C. After 30 min centrifugation at 16,000 *g* at 4 °C, the supernatant was discarded and the pellet was re-solubilized in 160 µL rehydration buffer (7 M urea, 2 M thiourea, 1% (wt/vol) DTT, 1% (wt/vol) CHAPS, 1% (wt/vol) Triton X-100, 1% (vol/vol) carrier ampholytes pH 3-10, trace amount of Bromophenol Blue). Fixed immobilized pre-cast IPG strips with a linear pH gradient 3–10 (cat. # ZM0018, Thermo Fisher Scientific, Waltham, MA, USA) were rehydrated in 155 µL of the resulting mixture overnight at room temperature inside IPG Runner cassettes (Thermo Fisher Scientific, Waltham, MA, USA). Isoelectrofocusing (first dimension separation) was performed at room temperature (175 V for 15 min, then 175–2000 V linear gradient for 45 min, then 2000 V for 30 min) on Life Technologies Zoom Dual Power Supply, using the XCell SureLock Mini-Cell Electrophoresis System (Thermo Fisher Scientific, Waltham, MA, USA). The IPG strips were then equilibrated for 15 minutes consecutively in (i) 6 M Urea, 20% (vol/vol) glycerol, 2% SDS, 375 mM Tris-HCl pH 8.8, 130 mM DTT and (ii) 6 M Urea, 20% (vol/vol) glycerol, 2% SDS, 375 mM Tris-HCl pH 8.8, 135 mM iodoacetamide, and loaded on 4–12% Bis-Tris ZOOM SDS-PAGE pre-cast gels (Thermo Fisher Scientific, Waltham, MA, USA). For the second dimension, SDS-PAGE was performed for 1 h at 170 V. Immunoblotting was performed as described elsewhere, blots were stained using ab3531 antibody (Abcam, Cambridge, MA, USA), visualized with FluorChem M FM 0512 imager and analyzed with AlfaView SA software (Protein Simple, San Jose, CA, USA).

For analysis of the hypersialylated versus hyposialylated isoform ratio, a previously published procedure was used [27]. Briefly, 2D blots were aligned horizontally and a line drawn at pI 7.5 was used to arbitrarily separate charge isoforms into hypersialylated and hyposialylated. In the software window, a rectangle was drawn to confine the spots of interest, and the densities were measured. The intensity of an equal background area from the same blot was subtracted before further analysis. The acquired spot ensemble intensities were used to calculate the ratio of hypersialylated isoforms over hyposialylated isoforms.

For visualizing intensities of individual sialoglycoforms on 2D, Alpha View software was used for converting the intensity of the dots within each 2D blot to a size of red spots, so that the intensity of the original signal is proportional to the size of red area. Such procedure enabled an assessment of relative intensities of individual sialoglycoforms within one gel. The threshold was manually adjusted for each 2D blot until the red-colored spots in all the gels were in the same size range.

### 4.5. qRT-PCR

Tissue pieces of about 20 µg from cortex, hippocampus, thalamus and stem were homogenized within 1.5 mL tubes in 200 µL Trizol (Thermo Fisher Scientific, Waltham, MA, USA), using RNase-free disposable pestles (Fisher scientific, Hampton, NH, USA). An additional 600 µL of Trizol was added, and the samples were centrifuged for 5 min at 12,000 *g*. Clear supernatant was transferred to a new tube and thoroughly mixed with 160 µL of cold chloroform. After 5 min incubation at room temperature, the samples were centrifuged for 15 min at 12,000 *g*, and the clear top layer was transferred to a new tube and mixed with an equal volume of cold 70% ethanol. The resulting mixture was further processed with Aurum Total RNA Mini Kit (Bio-Rad, Hercules, CA, USA) to isolate total RNA. To exclude genomic DNA contamination, the samples were treated with DNase I Total RNA, was dissolved in elution buffer and stored at -80 °C. An absorbance 260/280 value of ~2.0, determined using NanoDrop ND-1000 Spectrophotometer (Thermo Fisher Scientific, Waltham, MA, USA), proved RNA purity. Reverse transcription was performed using 1 μg of extracted RNA and iScript cDNA Synthesis Kit (Bio-Rad, Hercules, CA, USA). qRT-PCR was performed in triplicate from three normal and three prion-infected animals using SsoAdvanced Universal SYBR Green Supermix (Bio-Rad, Hercules, CA, USA) with Bio-Rad designed and validated primers: ST6Gal1 (qMmuCID0009827), ST6Gal2 (qMmuCED0046706), ST3Gal3 (qMmuCID0014977), ST3Gal4 (qMmuCID0007745) and ST3Gal6 (qMmuCED0046087). Housekeeping gene TBP (qMmuCID0040542) was used for normalization. The PCR protocol consisted of 95 °C for 2 min, followed by 40 amplification cycles with the following steps: 95 °C for 5 s, and 60 °C for 30 s. qRT-PCR was performed and analyzed using CFX96 Touch Real-Time PCR Detection System (Bio-Rad, Hercules, CA, USA), and plotted in Excel. 

## Figures and Tables

**Figure 1 ijms-21-00828-f001:**
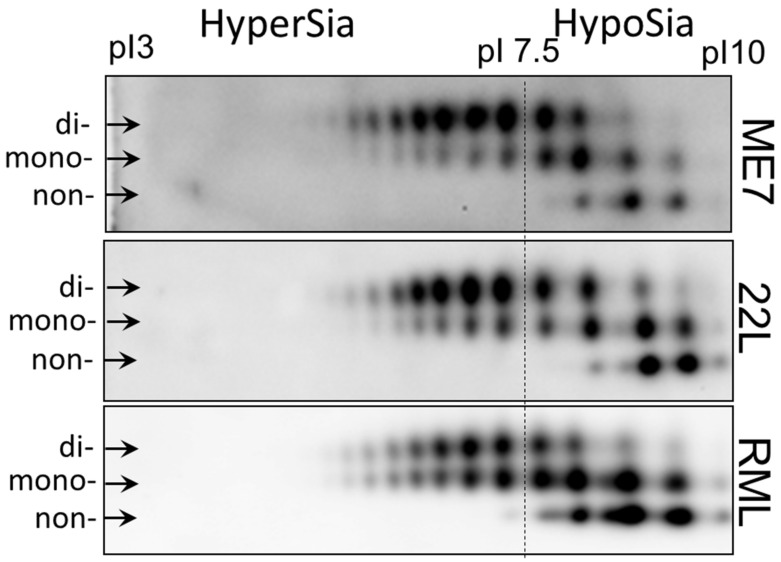
Analysis of strain-specific sialylation status of PrP^Sc^. Two-dimensional (2D) analysis of PrP^Sc^ from whole brain homogenate of animals infected with 22L, ME7 and RML. The samples were treated with PK. Anti-PrP antibody ab3531 were used for immunodetection. Arrows point at di-, mono- and non-glycosylated glycoforms. The dash line shows the position of pI 7.5 and arbitrary divides hypersialylated and hyposialylated PrP molecules. Appearance of more than one charge isoforms for non-glycosylated PrP is attributed to a structural heterogeneity of the GPI anchor [54]. As a result of N-glycan sialylation, the position of mono- and diglycosylated PrP isoforms is shifted toward acidic pH, in comparison non-glycosylated PrPs. Adapted from Katorcha et al. 2015 [27].

**Figure 2 ijms-21-00828-f002:**
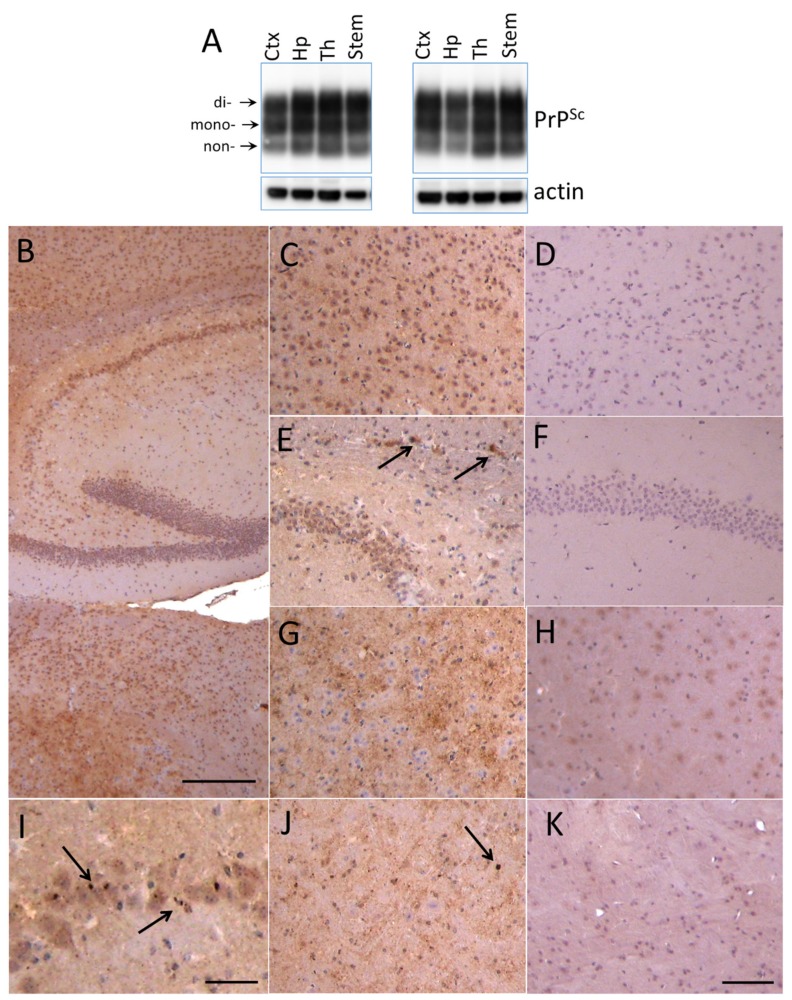
Deposition of 22L PrP^Sc^ in four brain regions. (**A**) Western blot analysis of 22L PrP^Sc^ deposition in cortex (Ctx), hippocampus (Hp), thalamus (Th) and stem of two terminally sick animals. Prior to Western blot of PrP^Sc^, brain material was treated with 20 µg/mL PK. Western blots were stained with anti-prion antibody ab3531. Arrows point at di-, mono- and non-glycosylated glycoforms. (**B**–**K**) Immunohistochemistry with SAF-84 showing PrP^Sc^ deposition in hippocampus (**B,E,I**), cortex (**C**), thalamus (**B,G**) and stem (**J**) of 22L-infected animals in comparison with corresponding regions of normal aged animals (**D,F,H,K**). Arrows focus on small plaques in the corpus callosum (**E**) and small granular deposits in pyramidal layer of hippocampus (**I**) and in stem (**J**). Scale bars: 300 µm on **B**, 100 µm on **C–H**, **J** and **K**, 50 µm on **I**.

**Figure 3 ijms-21-00828-f003:**
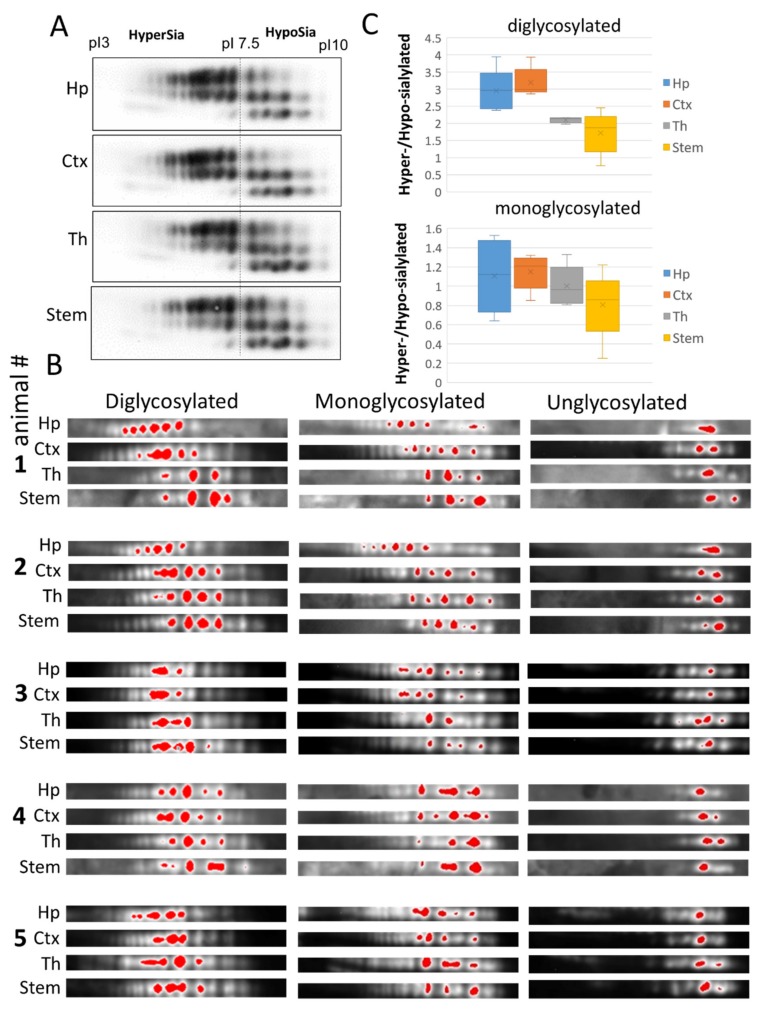
Analysis of sialylation status of PrP^Sc^ from four brain regions of 22L-infected animals. (**A**) Representative 2D Western blot of PrP^Sc^ from hippocampus (Hp), cortex (Ctx), thalamus (Th) and stem of 22L-infected animals. Prior to 2D blot, brain materials were treated with 20 µg/mL PK. (**B**) 2D Western blots of PrP^Sc^ from hippocampus (Hp), cortex (Ctx), thalamus (Th) and stem grouped according to PrP glycosylation status (di-, mono- and unglycosylated, *n =* 5 animals). Prior to 2D blot, brain materials were treated with with 20 µg/mL PK. Intensity of individual sialoglycoforms is visualized by red color using Alpha View software. (**C**) A box and whisker plot showing the ratio of hypersialylated versus hyposialylated PrP^Sc^ in hippocampus (Hp), cortex (Ctx), thalamus (Th) and stem. Diglycosylated and monoglycosylated sialoglycoforms are analyzed separately. The mean (x), the minimal and maximal values (the vertical line), and the medians of the bottom and the top half (the bottom and top lines of the box, respectively) are shown (*n* = 5 animals). The ratio of hypersialylated versus hyposialylated isoforms is calculated as described in Materials and Methods.

**Figure 4 ijms-21-00828-f004:**
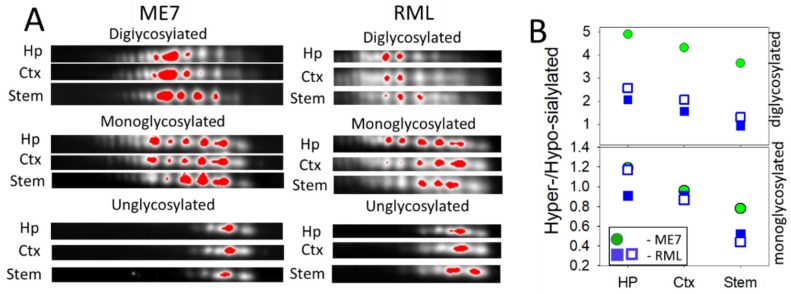
Analysis of sialylation status of PrP^Sc^ from ME7- and RML-infected animals. (**A**) 2D Western blot analysis of charge distribution of PrP^Sc^ sialoglycoforms from hippocampus (Hp), cortex (Ctx), and stem. Prior to 2D blot, brain materials were treated with 20 µg/ml PK. Intensity of individual sialoglycoforms is visualized by red color using Alpha View software. (**B**) A plot showing the ratio of hypersialylated versus hyposialylated PrP^Sc^ in hippocampus (Hp), cortex (Ctx) and stem in ME7-infected animal (circles, *n* = 1) and RML-infected animals (squares, *n =* 2). Diglycosylated and monoglycosylated sialoglycoforms are analyzed separately. The ratio of hypersialylated versus hyposialylated isoforms is calculated as described in Materials and Methods.

**Figure 5 ijms-21-00828-f005:**
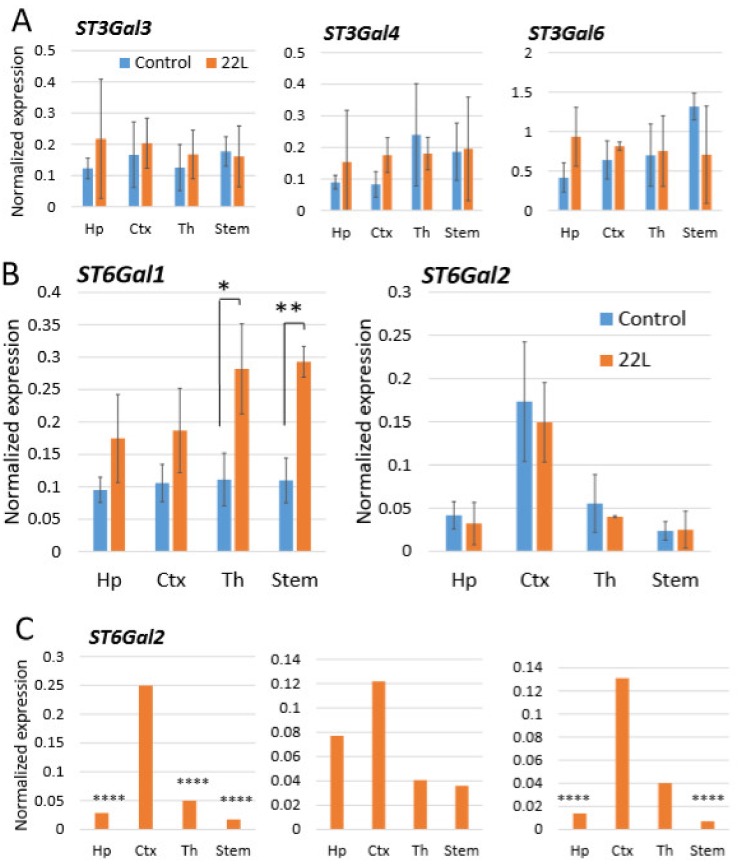
Analysis of gene expression by qRT-PCR. Expression of *ST3Gal3, ST3Gal4* and *ST3Gal6* (**A**), and *ST6Gal1* and *ST6Gal2* (**B**) in hippocampus (Hp), cortex (Ctx), thalamus (Th) and stem of 22L-infected animals or normal age-matched controls normalized by the expression of housekeeping gene *TBP*. The mean and standard deviation are shown (*n =* 3 animals). (**C**) Expression of *ST6Gal2* in hippocampus (Hp), cortex (Ctx), thalamus (Th) and stem of three individual 22L-infected animals normalized by the expression of housekeeping gene *TBP*. Statistical significance between the groups in (**A**) and (**B**) was calculated by Student’s unpaired t-test and indicated as * for *p* < 0.05 and ** for *p* < 0.01; in (**C**), **** indicates *p* < 0.0001 significance of differences (threshold regulation = 4) between samples analyzed in triplicate, and was calculated by BioRad CFX Manager.

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
