# Peer review of "Region-Specific Sialylation Pattern of Prion Strains Provides Novel Insight into Prion Neurotropism"

_ijms, 2020, doi:10.3390/ijms21030828_

Round 1

Reviewer 1 Report

In this report the authors attempt to access the role of prion sialylation in infectivity using three approaches (a) determining the distribution of sialylated form in infected animals with respect to strain, (b) determining relation of sialylation level to location of the infection in the brain and (c) measuring sialyltransferase expression in normal and infected brain tissues by region. The authors clearly show that the level of sialylation of the prion is related to infection, the prion strain and the region of the brain that is infected. However, while expression ST6Gal1 levels were higher in infected tissues, clear correlations between enzyme expression levels of STs and sialylation brain regions could not be determined. Thus, in the reviewer's opinion, the study is very interesting and should be published, but is not conclusive. 

Author Response

We thank the reviewer for his/her positive comments and appreciation the importance of this work.

Reviewer 2 Report

The manuscript reports the new observation in mouse models that the sialylation pattern of PrPSc differs in different region of brain.  The authors seek to shed light on the cause of the differential sialylation (putatively by differential expression of the sialyltransferase ST6GAL2), which is differentially expressed in different regions of the brain that was examined and correlated with the  PrPSc sialylation staus; similar trends are observed in three different prion strains (22L-, RML- and ME7-) infected brain tissues. Previous study from this group has demonstrated the importance of sialylation of PrPSc N-linked glycans in controlling prion fate and sialylation protects PrPSc against clearance and control the fate of prion infection.

The idea of regional-specificity of sialylation pattern of PrP and the possibility that this may be a disease driver is highly interesting and novel. The previous studies demonstrating that low sialylation status accelerates prion replication is an important premise for the current work.  This work, while interesting and is potentially highly significant, suffers the following drawback, which affects the overall impact and interpretation of the reported observation. The conclusion for regional specificity based on sialyltransferase expression differences is weak, and extensive discussion is needed to address alternative possibilities, which are not examined here.Potentially important alternatives are not considered in the manuscript, such as the possibility that there are differences in the underlying PrP glycan structures from the different brain regions, and that some of these PrP glycan forms are not open for further sialyltransferase modification.  Another alternative possibility is the substrate specificity for the sialyltransferases, and some forms of PrP glycosylation precursors are better targets for these sialyltransferases. Consideration for mass-spec based analysis of PrP recovered needs to be considered, or at least discussed. Much of the topic centers around heterogeneity of PrP-c and PrP-Sc glycosylation. A schematic representation of PrP depicting this will greatly improve the manuscript. Measurement of the sialyltransferase activity from the different brain regions may also bolster the claim that the pivotal checkpoint for differential PrP sialylation is in the sialyltransferase expression. It is well established that sialyltransferase expression on the mRNA level does not always equate to the level of sialyltransferase enzymatic activity.  Moreover, the acceptor-substrate specificity of ST6GAL2 is not well-studied; it is often believed that ST6GAL2 prefers small molecular weight acceptors, which PRP-glycans are definitely not. Additions points that should be addressed are as follows. These pertain to the overall poor quality of the writing, which can be easily improved for significantly highly impact and readability. The concept of “strains” in PrP is confusing for the non-prion reader. Please clarify and expand on the concept of “strain”: e.g. how strains are derived; how strains structurally differ; etc. Extremely confusing figures and legends Fig 1. Black, white triangles, arrows on left side of gels are extremely confusing. Please label them as di-, mono- and non-glycosylated on the figure itself. Fig 2. Panel A. If the nomenclature of triangles and arrows from Fig 1 is to be followed, please label the Fig 2 gels the same way. Fig 3. Difference between Panel A and Panel B not clear.  Both were described as 2D Western blots of PrP-SC from the same set of Hp, Ctx, Th, and stem.  Is the data in Panel A already in Panel B? Fig 3, Panel C. Y axis not labeled.  What are the numbers?  Presumably the numbers denote to the average sialic acid content? Fig 4 suffers from same issue. Is this set of data related to the same theme as Fig 3A, and how does this relate to Fig 3B?  is single ME7 and RML animal analysed, or is this representative animal of each?  For conclusion drawn that ranking order of PrP-Sc is strain-independent but brain region-dependent (Fig 4), it is not clear how many animals (N) of ME7 and RML were examined. Sufficient N should be examined to yield statistically-relevant conclusion. Regional differences in sialyltransferase expression may not be the only explanation for regional differences in PrP sialylation status. Other N-glycan structural differences, such as generation of bi- vs tetra-antennary N-glycan branches, leading to possibility of 2 vs 4 potential sialic acid accepting sites for each glycan, or the degree of galactosylation of each antennary termini. Penultimate galactose is required for sialyltransferase action.  In the case of the Fc N glycan on IgG, the low degree of Fc sialylation is largely attributed to the preponderance of under galactosylated Fc N-glycan, for example.  These other possibilities need to be considered at least in the Discussion.  Given the general audience, the glycoscience aspects need to be given a lay-person’s treatment.

Author Response

Comment 1. This work, while interesting and is potentially highly significant, suffers the following drawback, which affects the overall impact and interpretation of the reported observation. The conclusion for regional specificity based on sialyltransferase expression differences is weak, and extensive discussion is needed to address alternative possibilities, which are not examined here. Potentially important alternatives are not considered in the manuscript, such as the possibility that there are differences in the underlying PrP glycan structures from the different brain regions, and that some of these PrP glycan forms are not open for further sialyltransferase modification.  Another alternative possibility is the substrate specificity for the sialyltransferases, and some forms of PrP glycosylation precursors are better targets for these sialyltransferases. Consideration for mass-spec based analysis of PrP recovered needs to be considered, or at least discussed. Response. We agree that the data on region-specific expression of sialyltransferases are not conclusive. We rephrased our statements to avoid overinterpretation. Description of potential mechanisms that might account for region-specific differences in sialylation listed in this comment is added to the discussion of the revised manuscript.

Comment 2. Much of the topic centers around heterogeneity of PrP-c and PrP-Sc glycosylation. A schematic representation of PrP depicting this will improve the manuscript.  Response. Schematic representations of heterogeneity of PrP-c and PrP-Sc glycosylation  as well as selective recruitment of PrPC sialoglycoforms have been presented in several recent publications from our lab. We cite these publications, when needed, yet we prefer to avoid repetitive publishing of the schematic diagrams.

Comment 3. Measurement of the sialyltransferase activity from the different brain regions may also bolster the claim that the pivotal checkpoint for differential PrP sialylation is in the sialyltransferase expression. It is well established that sialyltransferase expression on the mRNA level does not always equate to the level of sialyltransferase enzymatic activity.  Moreover, the acceptor-substrate specificity of ST6GAL2 is not well-studied; it is often believed that ST6GAL2 prefers small molecular weight acceptors, which PRP-glycans are definitely not. Response. We appreciate this comment and understand the needs for measuring sialyltransferase activity. Unfortunately, this issue would be difficult to address experimentally within the current manuscript, because all work with prion-infected material is conducted in BSL-2+ facility, which currently does not have a set up for measuring sialyltransferase activity. The points that “sialyltransferase expression on the mRNA level does not always equate to the level of sialyltransferase enzymatic activity” and that “acceptor-substrate specificity of ST6GAL2 is not well-studied” are added to the discussion.

Comment 4. Additional points that should be addressed are as follows. These pertain to the overall poor quality of the writing, which can be easily improved for significantly highly impact and readability. The concept of “strains” in PrP is confusing for the non-prion reader. Please clarify and expand on the concept of “strain”: e.g. how strains are derived; how strains structurally differ; etc. Extremely confusing figures and legends Fig 1. Black, white triangles, arrows on left side of gels are extremely confusing. Please label them as di-, mono- and non-glycosylated on the figure itself. Fig 2. Panel A. If the nomenclature of triangles and arrows from Fig 1 is to be followed, please label the Fig 2 gels the same way. Response. The Introduction section is revised, and the introduction of the concept of prion strains is now expanded. Di-, mono- and non-glycosylated labels are placed in the figures 1 and 2. Figure legends are revised accordingly.

Comment 5. Fig 3. Difference between Panel A and Panel B not clear.  Both were described as 2D Western blots of PrP-SC from the same set of Hp, Ctx, Th, and stem.  Is the data in Panel A already in Panel B? Fig 3, Panel C. Y axis not labeled.  What are the numbers?  Presumably the numbers denote to the average sialic acid content? Fig 4 suffers from same issue. Is this set of data related to the same theme as Fig 3A, and how does this relate to Fig 3B?  is single ME7 and RML animal analysed, or is this representative animal of each?  For conclusion drawn that ranking order of PrP-Sc is strain-independent but brain region-dependent (Fig 4), it is not clear how many animals (N) of ME7 and RML were examined. Sufficient N should be examined to yield statistically-relevant conclusion.  Response. In Fig 3, panel A shows representative 2D blot for one animal. Panel B shows 2D of PrPSc from five animals (including the animal from the panel A), yet the data are organized in a different manner: strips corresponding to di-, mono-, and unglycosylated sialoglycoforms are shown for four brain regions. Panel C: the Y-axis was labeled as Hyper/Hyper-sialylated in the plot and was also indicated in the figure legend.  The procedure for calculating Hyper/Hyper-sialylated ratio was described in Materials and Methods. In Fig. 4, one ME7- and two RML-infected animals were examined (n is now included in figure legend). Currently, we do not have fresh ME7 and RML brain materials for examining more animals for these strains; their preparation would take one year. Statements that statistically sound differences between brain regions could not be established for RML and ME7 animals are added to Result and Discussion. Our conclusion on region-dependent ranking order of PrPSc sialylation is based on analysis of 22L brains (N=5). New panel B in figure 4 illustrate that ME7 and RML follow the same trend as 22L. We apologize for the misunderstanding of our conclusions regarding the role of strain versus brain region. Prion strains do show strain-dependent sialylation patterns of PrPSc (See Fig.1). As stated in the discussion “….in addition to strain-specific differences, PrPSc sialylation pattern is controlled by a brain region.”

Comment 6. Regional differences in sialyltransferase expression may not be the only explanation for regional differences in PrP sialylation status. Other N-glycan structural differences, such as generation of bi- vs tetra-antennary N-glycan branches, leading to possibility of 2 vs 4 potential sialic acid accepting sites for each glycan, or the degree of galactosylation of each antennary termini. Penultimate galactose is required for sialyltransferase action.  In the case of the Fc N glycan on IgG, the low degree of Fc sialylation is largely attributed to the preponderance of under galactosylated Fc N-glycan, for example.  These other possibilities need to be considered at least in the Discussion.  Given the general audience, the glycoscience aspects need to be given a lay-person’s treatment. Response. Alternative possibilities such as region-specific structural differences of N-glycan are now discussed.

Reviewer 3 Report

The study by Makarava is a logical extension to previous work by the Baskakov laboratory that explored the relationship between PrP sialylation state and prion conversion. Here, the authors asked whether the brain region-specific level of sialylation influences the regional susceptibility to prion conversion, using three well-established prion strains (22L, ME7, RML) and mice as their study object. The work relied on sophisticated and technically challenging methods for the analysis of PrP sialylation, and the data are mostly well-presented.

There is currently no other group that would be better equipped to address this interesting question and I commend the team for taking their analysis to this level. I consider this work of outstanding interest to the field and recommend publication once a few smaller points have been adressed.

1. Fig 3 documenting the detailed comparison of 22L PrPSc in brain regions of several mice is highly instructive and useful for establishing the degree of similarity and variation observed. The information in the related Fig. 4, showing the same type of information for mice infected with RML and ME7 prions, would be easier to grasp if it was connected back to Fig. 3. Appreciating the considerable effort involved in generating the data underlying these figures, this is not a request to generate more data but to introduce some graph that compares the levels of PrPSc sialylation across the three prion strains and three tissues that were consistently evaluated in the subpanels shown in Fig. 3 and 4.

2. Lanes 228-229: The authors state: "Observation of region-specific sialylation would indicate that a link between PrPSc 228 sialylation and differential vulnerability exist." Although a body of work that has come out of the Baskakov group over the past few years has established a relationship between sialylation state and PrP conversion characteristics, a causal in vivo link is, to the best of my knowledge, still missing (cited work by the Manson group is suggestive but also not conclusive) and was also not established in this work, which was correlative in its nature. The authors are mostly careful when interpreting their findings but may want to also rephrase the sentence above to better reflect this.

3. Lane 258-260: The authors state "It would be interesting to test the effect of ST6Gal2 knock out on PrPSc sialylation and prion propagation. Unfortunately, despite multiple attempts, the efforts to generate ST6Gal2 knock out mice have not been successful so far."
Any reader of this manuscript will want to understand what exactly is holding up such analysis, given that it would be best-suited to establish a causal relationship. The authors should qualify this statement by clarifying the nature of the attempts undertaken thus far and the presumed reasons for why they have been unsuccessful.

4. In a subset of figure panels, legends are too short, making it a bit cumbersome to understand what is presented, e.g., lacking are descriptions on whether PrPSc bands represented PK digested material or where assigned otherwise (e.g., through PrPSc-specific antibodies.

5. Lane 149: "For visualizing the difference in more details, we applied artificial color assignment using Alpha View software".
Please elaborate a bit more on the filtering applied that underlay red-colored signals. Were the same filters applied consistently or did thresholds differ between analyses?

6. Lane 319: Fix title 'D electrophoresis'

Author Response

Comment 1. Fig 3 documenting the detailed comparison of 22L PrPSc in brain regions of several mice is highly instructive and useful for establishing the degree of similarity and variation observed. The information in the related Fig. 4, showing the same type of information for mice infected with RML and ME7 prions, would be easier to grasp if it was connected back to Fig. 3. Appreciating the considerable effort involved in generating the data underlying these figures, this is not a request to generate more data but to introduce some graph that compares the levels of PrPSc sialylation across the three prion strains and three tissues that were consistently evaluated in the subpanels shown in Fig. 3 and 4. Response.  In response to this comments, the values for RML and ME7 are now plotted in new panel B, Fig. 4. The same trends were observed for RML and 22L, as those reported for 22L in Fig. 3.

Comment 2. Lanes 228-229: The authors state: "Observation of region-specific sialylation would indicate that a link between PrPSc sialylation and differential vulnerability exist." Although a body of work that has come out of the Baskakov group over the past few years has established a relationship between sialylation state and PrP conversion characteristics, a causal in vivo link is, to the best of my knowledge, still missing (cited work by the Manson group is suggestive but also not conclusive) and was also not established in this work, which was correlative in its nature. The authors are mostly careful when interpreting their findings but may want to also rephrase the sentence above to better reflect this. Response. This statement was rephrased to “…is consistent with the hypothesis that a link between PrPSc sialylation and differential vulnerability exist

Comment 3. Lane 258-260: The authors state "It would be interesting to test the effect of ST6Gal2 knock out on PrPSc sialylation and prion propagation. Unfortunately, despite multiple attempts, the efforts to generate ST6Gal2 knock out mice have not been successful so far."
Any reader of this manuscript will want to understand what exactly is holding up such analysis, given that it would be best-suited to establish a causal relationship. The authors should qualify this statement by clarifying the nature of the attempts undertaken thus far and the presumed reasons for why they have been unsuccessful. Response. This statement was based on personal communications with several experts in glycobiology, who attempted to generate ST6Gal2 KO in a past, but without success. Nevertheless, we just learned that global ST6Gal2 KO has been developed recently using CRISPR. Aforementioned sentence is deleted in revised manuscript.

Comment 4.  In a subset of figure panels, legends are too short, making it a bit cumbersome to understand what is presented, e.g., lacking are descriptions on whether PrPSc bands represented PK digested material or where assigned otherwise (e.g., through PrPSc-specific antibodies. Response. We apologize for omitting this information. Figure legends are revised.

Comment 5.  Lane 149: "For visualizing the difference in more details, we applied artificial color assignment using Alpha View software". Please elaborate a bit more on the filtering applied that underlay red-colored signals. Were the same filters applied consistently or did thresholds differ between analyses? Response.  With this feature, the intensity of the dots within one 2D gel was proportional to the size of the red spots, enabling an assessment of relative intensities within one gel. For the comparison of different gels, the threshold was manually adjusted until the red-colored spots in all the gels were in the same size range. This procedure is now described in Material and Methods.

Comment 6.  Lane 319: Fix title 'D electrophoresis'. Response. Fixed.